# Synergistic Approach for Simultaneous Optimization of Monolingual, Cross-lingual, and Multilingual Information Retrieval

## Abstract

Information retrieval across different languages is an increasingly important challenge in natural language processing. Recent approaches based on multilingual pre-trained language models have achieved remarkable success, yet they often optimize for either monolingual, cross-lingual, or multilingual retrieval performance at the expense of others. This paper proposes a novel hybrid batch training strategy to simultaneously improve zero-shot retrieval performance across monolingual, cross-lingual, and multilingual settings while mitigating language bias. The approach fine-tunes multilingual language models using a mix of monolingual and cross-lingual question-answer pair batches sampled based on dataset size. Experiments on XQuAD-R, MLQA-R, and MIRACL benchmark datasets show that the proposed method consistently achieves comparable or superior results in zero-shot retrieval across various languages and retrieval tasks compared to monolingual-only or cross-lingual-only training. Hybrid batch training also substantially reduces language bias in multilingual retrieval compared to monolingual training. These results demonstrate the effectiveness of the proposed approach for learning language-agnostic representations that enable strong zero-shot retrieval performance across diverse languages.

## 1 Introduction

Information retrieval (IR) across different languages is an increasingly important challenge in natural language processing. However, optimizing information retrieval systems for multilingual scenarios is not a straightforward task, as it requires considering multiple distinct retrieval settings, each with its own set of challenges and requirements, including monolingual retrieval, cross-lingual retrieval, and multilingual retrieval. Monolingual retrieval refers to the task of retrieving documents in the same language as the user's query, focusing on developing effective ranking algorithms and relevance matching techniques. Cross-lingual retrieval involves queries and documents in different languages, requiring the system to bridge the language gap by employing techniques such as query translation, document translation, or cross-lingual representation learning. Multilingual retrieval requires the creation of a single ranked list of documents in multiple languages for a given query, addressing challenges such as language disparity, varying document lengths, and potential differences in content quality and relevance across languages while providing users with a unified and coherent ranked list of results.

Recent approaches to multilingual information retrieval have leveraged multilingual pre-trained language models such as mBERT (Devlin et al., 2019) and XLM-R (Conneau et al., 2020) to encode queries and documents (Karpukhin et al., 2020). While these models can transfer relevance matching capabilities across languages, their performance tends to underperform on cross-lingual retrieval benchmarks due to the lack of explicit alignment between languages during pretraining (Zhang et al., 2023). LaREQA, introduced by (Roy et al., 2020), targets strong alignment, requiring semantically related pairs across languages to be closer in representation space than unrelated pairs within the same language. (Roy et al., 2020) found that augmenting the training data through machine translation proved effective in achieving robust alignment for MLIR. However, this approach compromises performance in monolingual retrieval tasks. Alternative approaches using parallel corpora, such as InfoXLM (Chi et al., 2021) and LaBSE (Feng et al., 2022), have been proposed to align sentences

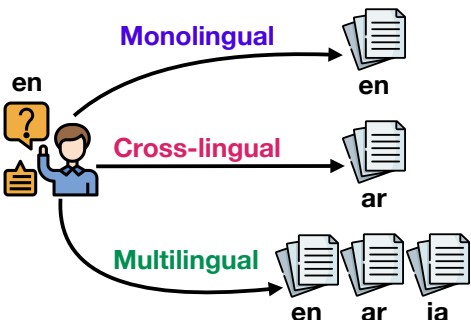

Figure 1: Illustrative example of monolingual, cross-lingual, and multilingual information retrieval.

across languages. However, the scarcity of parallel data, especially for low-resource languages, remains a substantial challenge. To address these limitations, (Lawrie et al., 2023) introduced a Multilingual Translate-Train approach using translated datasets, (Hu et al., 2023) proposed contrastive losses to align representations and remove language-specific information, (Huang et al., 2023a) presented a knowledge distillation framework for multilingual dense retrieval, and (Lin et al., 2023a) extended Aggretriever (Lin et al., 2023b) for multilingual retrieval using semantic and lexical features. While the methods proposed in (Hu et al., 2023) and (Huang et al., 2023a) attempt to mitigate language bias, we raise the question: Is there a straightforward approach that addresses this issue by modifying the training data batches without necessitating the introduction of loss functions or new architectural components?

In this paper, we propose a novel hybrid batch training strategy that simultaneously optimizes retrieval performance across monolingual, cross-lingual, and multilingual settings while also mitigating language bias. Our approach fine-tunes multilingual language models using a balanced mix of monolingual and cross-lingual question-answer pair batches. We collect a diverse set of English question-answer datasets and use machine translation to generate parallel question-answer pairs across several languages, including low-resource languages where parallel corpora may be limited (Fan et al., 2021; Kim et al., 2021; Costa-jussà et al., 2022). Our hybrid batch training approach significantly reduces the language bias that hinders the performance of multilingual retrieval systems by training the models on a diverse set of language pairs and encouraging the learning of language-agnostic representations. This mitigates the tendency of models to favor certain languages over others, ensuring that documents from multiple languages are fairly ranked based on their relevance to the query, regardless of the language. Extensive experiments on XQuAD-R, MLQA-R, and MIRACL benchmark datasets demonstrate the effectiveness of our proposed approach, with models trained using the hybrid batch strategy consistently achieving competitive results in zero-shot retrieval across various languages and retrieval tasks, outperforming models trained with only monolingual or cross-lingual data. Our approach also exhibits strong zero-shot generalization to unseen languages not included in the training data, highlighting its potential to expand the linguistic coverage of multilingual information retrieval systems.

## 2 METHODOLOGY

### 2.1 CONTRASTIVE LEARNING

Throughout the paper, we utilize the dual-encoder architecture with shared parameters, which is commonly used for dense retrieval (DR; Ni et al., 2022). Contrastive learning is a method for training DR models by contrasting positive pairs against negatives. Specifically, given a batch of triplets, each of which consists of a query and its relevant and irrelevant documents: $(q_n, d_n^+, d_n^-); 1 \le n \le |\mathbf{B}|$. We minimize the InfoNCE loss for each query $q_n$:

$$\mathcal{L} = \sum_{i=1}^{|\mathbf{B}|} - \log \frac{e^{s_\theta(q_i, d_i^+)}}{e^{s_\theta(q_i, d_i^+)} + \sum_{j=1}^{|\mathbf{B}|} e^{s_\theta(q_i, d_j^-)}}. \tag{1}$$

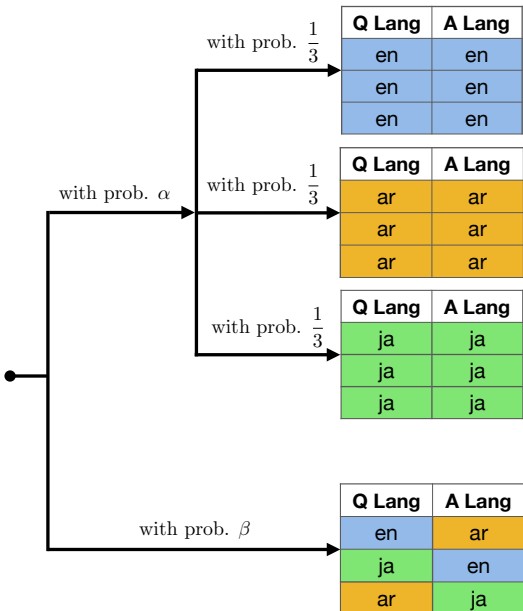

(a) Proposed hybrid batching

Figure 2: Illustrations of the proposed hybrid batch sampling (assuming we only have training data in English, Arabic, and Japanese), where our model is exposed to monolingual and cross-lingual batches with the respective probability of $\alpha$ and $\beta = 1 - \alpha$.

We use cosine similarity as the scoring function: $s_\theta(q, d) = \cos(\mathbf{E}_\theta(q), \mathbf{E}_\theta(d))$, where $\mathbf{E}_\theta$ is the encoder parametrized by $\theta$. Following Wang et al. (2022), we incorporate prefix identifiers "`Query:`" and "`Passage:`" for queries and passages, respectively. As shown in prior work (Hofstätter et al., 2021; Lin et al., 2021), in-batch negatives mining, the second term of the denominator in Eq (1), plays a crucial role in dense retrieval training. In this work, we study different batch sampling approaches to control in-batch negative mining.

## 2.2 BATCH SAMPLING

**Baseline Batch Sampling.** We study the following training batching procedures introduced by (Roy et al., 2020). (i) Monolingual batching (coined as X-X-mono model) creates each batch with mono language, where all the triplets consist of queries and passages in the same language. Note that we sample the language used to create the batch equally among all possible languages in our training data. (ii) Cross-lingual batching (coined as X-Y model) creates each batch, where all the triplets consist of queries and passages in different languages. Monolingual batching only focuses on contrastive learning for query-passage pairs in the same languages while cross-lingual batching mines positives and in-batch negatives from diverse languages.

As shown in (Roy et al., 2020), the X-Y model is more effective in cross-lingual retrieval scenarios and shows reduced language bias; however, the X-X-mono surpasses the X-Y model in monolingual retrieval. These results inspire us to explore whether simply combining the two batch sampling approaches can achieve improvement in both monolingual and cross-lingual retrieval effectiveness.

**Hybrid Batch Sampling.** In this work, we propose to combine the two aforementioned baseline sampling strategies. Specifically, when creating batch training data, we set $\alpha$ and $\beta = 1 - \alpha$ as the respective probability of using monolingual and cross-lingual batching as shown in Fig. 2.[1]

---

[1]In the experiments, we found out that setting the hyperparameters $\alpha$ and $\beta$ to 0.5 resulted in the best balance between the performance of the proposed model on monolingual and multilingual evaluations.

## 3 EXPERIMENTAL SETUP

This section presents the experimental setup for evaluating the proposed hybrid batch training strategy. We first discuss the training process, including datasets, and multilingual pre-trained models. Next, we introduce the evaluation datasets and metrics used to assess the performance of the fine-tuned models. Finally, we describe the evaluation settings for monolingual, cross-lingual, and multilingual retrieval tasks.

### 3.1 TRAINING

**Datasets.** To conduct the study of batch sampling, parallel query-passage training pairs are required such that we can construct cross-lingual triplets, where each query and its relevant (or irrelevant) passage are in different languages. mMARCO (Bonifacio et al., 2021) is the only dataset with parallel queries and passages across 14 languages. In our study, we further scale the size of training data by translating the existing question-answering datasets. Specifically, we developed our in-house machine translation pipeline to create parallel QA pairs for the monolingual datasets across nine languages: Arabic, Chinese, English, German, Hindi, Russian, Spanish, Thai, and Turkish. The additional training data used in our study include DuoRC (Saha et al., 2018), EntityQuestions (Sciavolino et al., 2021), Google NQ (Kwiatkowski et al., 2019), MFAQ (De Bruyn et al., 2021), Mr. Tydi (Zhang et al., 2021), NewsQA (Trischler et al., 2017), WikiQA (Yang et al., 2015), and Yahoo QA mined from Yahoo Answers. Appendix A.1 provides comprehensive details about the training datasets.

**Training Setup.** We apply the baseline and our proposed hybrid batching to fine-tune two representative multilingual pre-trained models: (i) XLM-RoBERTa (XLM-R) (Conneau et al., 2020); and (ii) language-agnostic BERT sentence embedding (LaBSE) (Feng et al., 2022). Model training experiments were conducted using one NVIDIA A100-80 GB GPU. We fine-tune pre-trained models using AdamW optimizer (Loshchilov & Hutter, 2018) with weight decay set to 1e-2, a learning rate of 3e-5, and a batch size of 100. We apply the early stopping (Prechelt, 1998) to select the model checkpoint with the lowest validation loss on SQuADShifts dataset (Miller et al., 2020). Note that the validation set used for checkpoint selection consists solely of English examples.

**Hyperparameter Tuning for Hybrid Batch Sampling.** To determine the optimal values for the hyperparameters $\alpha$ and $\beta$ in our hybrid batch sampling approach, we conducted a comprehensive grid search. We evaluated $\alpha$ values ranging from 0 to 1, with $\beta$ always set to $1 - \alpha$. Each configuration was tested on a held-out validation set comprising a diverse selection of languages. We assessed the model's performance across monolingual, cross-lingual, and multilingual retrieval tasks. Our goal was to find a balance that would optimize performance across all three retrieval settings without significantly sacrificing any particular one. We found that setting $\alpha = 0.5$ provided the best overall results, striking an effective balance between monolingual and cross-lingual/multilingual performance. This equal weighting between monolingual and cross-lingual batches allowed our model to maintain strong monolingual retrieval capabilities while also excelling in cross-lingual and multilingual scenarios. We also observed that the model's performance was relatively stable for $\alpha$ values between 0.4 and 0.6, indicating some robustness to small variations in these hyperparameters.

### 3.2 EVALUATION

**Datasets.** We evaluate the retrieval effectiveness of different models on three distinct datasets: XQuAD-R (Roy et al., 2020) and MLQA-R (Roy et al., 2020).[2] XQuAD-R and MLQA-R are question-answering datasets with parallel questions and passages in 11 languages and 7 languages, respectively. Thus, these two datasets can be used to evaluate monolingual, cross-lingual, and multilingual retrieval effectiveness. Appendix A.2 provides comprehensive details about the evaluation datasets. Furthermore, we report the detailed monolingual retrieval effectiveness on MIRACL dev (Zhang et al., 2022) in Table 12 and 13 in Appendix A.3.1.

---

[2]The evaluation of the models is conducted on datasets that are completely separate and distinct from the ones used for training. More specifically, the models have not encountered any data samples, whether from the training or testing splits, of the evaluation datasets during their training process. This ensures an unbiased assessment of the ability of the models to generalize and perform effectively on unseen data.

Table 1: Main experiments on XQuAD-R and MLQA-R. mAP (marco averaged across all languages) numbers are reported. Mo., CR., and Mul. denote monolingual, cross-lingual, and multilingual retrieval settings. respectively.

| Model | Sampling | XQuAD-R (↑) | | | MLQA-R (↑) | | |
|---|---|---|---|---|---|---|---|
| | | Mo. | Cr. | Mul. | Mo. | Cr. | Mul. |
| XLM-R | X-X | .792 | .674 | .547 | .648 | .584 | .473 |
| | X-Y | .755 | .700 | **.593** | .626 | .620 | .508 |
| | Hybrid | **.798** | **.705** | .593 | **.648** | **.623** | **.512** |
| LaBSE | X-X | .808 | .752 | .652 | .681 | .656 | .550 |
| | X-Y | .801 | .762 | .679 | .671 | .677 | .576 |
| | Hybrid | **.817** | **.767** | **.682** | **.686** | **.681** | **.579** |

Table 2: Language bias in multilingual retrieval.

| Model | Sampling | language bias (↓) | |
|---|---|---|---|
| | | XQuAD-R | MLQA-R |
| XLM-R | X-X | 410 | 288 |
| | X-Y | 295 | **227** |
| | Hybrid | **287** | 227 |
| LaBSE | X-X | 262 | 225 |
| | X-Y | 225 | 198 |
| | Hybrid | **221** | **195** |

**Metrics and Settings.** We report the mean average precision (mAP) for XQuAD-R and MLQA-R since the metric considers the retrieval quality when multiple relevant passages for a given query exist.[3] We conduct retrieval using the queries with $X_Q$ language against the corpus with $X_C$ language and report the macro-averaged mAP over all the cross-lingual (denoting Cr.) combinations language pairs ($X_Q \neq X_C$), and the other monolingual (denoting Mo.) combinations ($X_Q = X_C$). For example, in XQuAD-R (MLQA-R), we have 11 and 7 parallel languages; thus, there are 110 (42) and 11 (7) cross-lingual and monolingual retrieval settings, respectively. For multilingual (denoting Mul.) retrieval, we conduct retrieval using the queries with $X_Q$ language against all the parallel corpus in different languages. We report the detailed results for specific languages in Section 4.2.

## 4 EXPERIMENTAL RESULTS

### 4.1 SUMMARY OF MAIN RESULTS

**Zero-shot Retrieval Evaluation.** We report the effectiveness of different batch sampling strategies in Table 1. We observe that X-X and X-Y sampling only perform well in monolingual and cross-lingual retrieval settings, respectively. These results indicate that optimization for either monolingual or cross-lingual retrieval alone may come at the expense of the other. Our hybrid batch sampling, on the other hand, optimizes both retrieval settings. As a result, our hybrid batch sampling achieves the best performance in multilingual retrieval settings, where the ability of the models to handle both monolingual and cross-lingual retrieval tasks is evaluated.[4] Finally, the same conclusion holds when using XLM-R and LaBSE as initialization that hybrid batch sampling is better than the other two baseline batch sampling approaches. A thorough analysis of the retrieval performance across various training batch types, retrieval tasks, languages, and datasets is presented in Section 4.2.1.

---

[3]The results for the Recall metric are in Section 4.2.1.

[4]The performance of the models is evaluated on certain languages, such as Greek (el) and Vietnamese (vi), which were not included in the training data. This aspect of the evaluation process aims to assess the ability of the models to handle languages they have not been explicitly trained on, providing insights into their zero-shot cross-lingual transfer capabilities (See Section 4.2.1).

In particular, Tables 3 through 6 showcase the MAP and Recall scores for zero-shot monolingual, cross-lingual, and multilingual retrieval tasks on the XQuAD-R and MLQA-R datasets, considering both fine-tuned XLM-R and LaBSE models.

**Language Bias Evaluation.** To gain insight into why hybrid batch sampling achieves strong performance in multilingual retrieval settings, we investigate the language bias exhibited by models fine-tuned using different batch sampling strategies. Following Huang et al. (2023b), we measure the language bias using the maximum rank distance among all the parallel corpus. That is, for each query, we calculate the difference between the highest and lowest rank of the relevant passages.[5] We report the macro averaged rank distance across all languages in Table 2 and present the comprehensive results in Section 4.2.2. Specifically, Table 7 shows the rank distances for the XQuAD-R dataset, while Table 8 displays the rank distances for the MLQA-R dataset, both considering fine-tuned XLM-R and LaBSE models under different training batch types. As shown in Table 2, models fine-tuned with cross-lingual batch sampling show less language bias compared to those fine-tuned with multi-lingual batch sampling. It is worth noting that our hybrid batch sampling, combining both baseline sampling, still maintains low language bias without sacrificing monolingual retrieval effectiveness.

## 4.2 In-depth Analysis

### 4.2.1 Zero-shot Retrieval Evaluation on XQuAD-R and MLQA-R

We present the experimental results of our proposed hybrid batching approach for improving the retrieval performance of fine-tuned multilingual language models across various tasks and datasets. We compare our method with two baseline training batch methods (X-X-mono and X-Y) using two pre-trained multilingual language models (XLM-R and LaBSE) on two evaluation datasets (XQuAD-R and MLQA-R). The performance is measured using Mean Average Precision (MAP) and Recall @ 1 (R@1) and Recall @ 10 (R@10) metrics across monolingual, cross-lingual, and multilingual retrieval settings.

**Consistent improvement across languages and tasks:** Tables 3 through 6 demonstrate the performance of the proposed hybrid batching approach when applied to the XLM-R and LaBSE models on the XQuAD-R and MLQA-R datasets. Our method consistently achieves the highest mean MAP and mean R@1 scores across monolingual and cross-lingual settings for all combinations of datasets and models. Furthermore, our proposed method consistently achieves either the highest mean MAP and mean R@10 scores in the multilingual retrieval setting or performs comparably to the X-Y batching method, which is specifically optimized for multilingual retrieval. Notably, there is a substantial performance gap between the second-best approach (either our method or X-Y) and the third-best approach (X-X-mono) in terms of these evaluation metrics for multilingual retrieval. This demonstrates the robustness and effectiveness of the proposed method in improving retrieval performance, regardless of the language or task complexity.

**Balanced performance across evaluation metrics:** The proposed approach strikes a balance between the X-X-mono (optimized for monolingual retrieval setting) and X-Y (cross-lingual/multilingual retrieval settings) baselines. This compromise is evident when analyzing the performance of individual languages across different retrieval tasks. In the monolingual retrieval setting, the proposed method tends to outperform or maintain comparable performance to the X-X-mono baseline for most languages. Similarly, the proposed approach generally surpasses the X-Y baseline across most languages in the cross-lingual and multilingual retrieval settings. A key insight is that in cases where our approach does not achieve the top performance for a specific language and retrieval setting, it consistently performs as a strong runner-up to the approach specifically optimized for that retrieval setting. Simultaneously, our method maintains a significant advantage over the third-best approach in such cases. This trend is consistent for XLM-R and LaBSE models on the XQuAD-R and MLQA-R datasets. By effectively finding a middle ground between the strengths of the X-X-mono and X-Y baselines, the proposed method offers a versatile solution that can handle monolingual, cross-lingual, and multilingual retrieval tasks across a wide range of languages without significantly compromising performance in any particular setting.

---

[5]Note that in XQuAD-R and MLQA-R, each query only has one relevant passage in each language.

Table 3: Performance comparison of MAP and Recall scores across zero-shot monolingual, cross-lingual, and multilingual retrieval tasks on the XQuAD-R dataset for a fine-tuned XLM-R model and different training batch types. The best result is highlighted in **bold**, and the second-best result is underlined.

| | Evaluation of Fine-tuned XLM-R Model on XQuAD-R Dataset | | | | | | | | |
|---|---|---|---|---|---|---|---|---|---|
| | MAP | | | | | | | | |
| | Monolingual | | | Cross-lingual | | | Multilingual | | |
| Source Language | X-X-mono | X-Y | Proposed | X-X-mono | X-Y | Proposed | X-X-mono | X-Y | Proposed |
| ar | 0.7581 | 0.7318 | **0.7619** | 0.6064 | **0.6607** | 0.6564 | 0.487 | **0.5519** | 0.5416 |
| de | 0.7893 | 0.7694 | **0.8033** | 0.6979 | 0.7147 | **0.7222** | 0.5653 | 0.6113 | **0.6133** |
| el | 0.7749 | 0.7226 | **0.7844** | 0.6492 | 0.6791 | **0.683** | 0.5127 | **0.5638** | 0.5599 |
| en | 0.8327 | 0.7892 | **0.8389** | 0.7247 | 0.7319 | **0.7473** | 0.5984 | 0.631 | **0.6436** |
| es | 0.8019 | 0.7617 | **0.8089** | 0.7072 | 0.7178 | **0.7332** | 0.582 | 0.6123 | **0.6245** |
| hi | 0.778 | 0.7461 | **0.787** | 0.641 | **0.6835** | 0.676 | 0.5171 | **0.5787** | 0.5666 |
| ru | 0.802 | 0.7758 | **0.8125** | 0.694 | 0.7103 | **0.7186** | 0.5763 | 0.6076 | **0.6104** |
| th | 0.7634 | 0.7312 | **0.7697** | 0.6623 | 0.6963 | **0.6978** | 0.5442 | 0.5862 | **0.5876** |
| tr | 0.7801 | 0.7479 | **0.7913** | 0.6748 | 0.7013 | **0.7078** | 0.5524 | **0.6005** | 0.5989 |
| vi | **0.8113** | 0.7624 | 0.8025 | 0.6742 | 0.6904 | **0.7017** | 0.5417 | **0.5817** | 0.5781 |
| zh | **0.8178** | 0.771 | 0.8146 | 0.6795 | 0.7105 | **0.7144** | 0.5496 | **0.6023** | 0.5957 |
| Mean | 0.7918 | 0.7554 | **0.7977** | 0.6737 | 0.6997 | **0.7053** | 0.5479 | **0.5934** | 0.5927 |
| | R@1 | | | | | | R@10 | | |
| | Monolingual | | | Cross-lingual | | | Multilingual | | |
| Source Language | X-X-mono | X-Y | Proposed | X-X-mono | X-Y | Proposed | X-X-mono | X-Y | Proposed |
| ar | 0.6596 | 0.6276 | **0.6639** | 0.4907 | **0.5463** | 0.5419 | 0.4272 | **0.4811** | 0.4722 |
| de | 0.698 | 0.6726 | **0.7149** | 0.5883 | 0.6053 | **0.6148** | 0.4929 | 0.5308 | **0.5322** |
| el | 0.6875 | 0.6166 | **0.6968** | 0.531 | 0.5666 | **0.5726** | 0.4495 | 0.4904 | **0.4923** |
| en | 0.7523 | 0.6942 | **0.7582** | 0.62 | 0.6246 | **0.6447** | 0.5196 | 0.5445 | **0.5594** |
| es | 0.7207 | 0.6624 | **0.7232** | 0.5986 | 0.6096 | **0.6287** | 0.5067 | 0.5303 | **0.5439** |
| hi | 0.6881 | 0.6517 | **0.6999** | 0.5276 | **0.574** | 0.5664 | 0.4514 | **0.5043** | 0.4957 |
| ru | 0.7108 | 0.6788 | **0.7277** | 0.5848 | 0.5994 | **0.6115** | 0.5047 | 0.5299 | **0.5323** |
| th | 0.6703 | 0.6272 | **0.6729** | 0.5481 | **0.5875** | 0.5871 | 0.4781 | 0.5127 | **0.5141** |
| tr | 0.69 | 0.6453 | **0.6959** | 0.5669 | 0.5932 | **0.6026** | 0.4825 | 0.5196 | **0.5219** |
| vi | **0.7301** | 0.6599 | 0.7132 | 0.5631 | 0.5798 | **0.5949** | 0.4703 | **0.5038** | 0.5015 |
| zh | **0.7307** | 0.6732 | 0.7282 | 0.5666 | 0.6011 | **0.6081** | 0.4806 | **0.523** | 0.5208 |
| Mean | 0.7035 | 0.6554 | **0.7086** | 0.5623 | 0.5898 | **0.5976** | 0.4785 | 0.5155 | **0.5169** |

Table 4: Performance comparison of MAP and Recall scores across zero-shot monolingual, cross-lingual, and multilingual retrieval tasks on the MLQA-R dataset for a fine-tuned XLM-R model and different training batch types. The best result is highlighted in **bold**, and the second-best result is underlined.

| | Evaluation of Fine-tuned XLM-R Model on MLQA-R Dataset | | | | | | | | |
|---|---|---|---|---|---|---|---|---|---|
| | MAP | | | | | | | | |
| | Monolingual | | | Cross-lingual | | | Multilingual | | |
| Source Language | X-X-mono | X-Y | Proposed | X-X-mono | X-Y | Proposed | X-X-mono | X-Y | Proposed |
| ar | 0.5973 | 0.577 | **0.6006** | 0.5351 | **0.5837** | 0.5787 | 0.4091 | 0.456 | **0.4602** |
| de | 0.5915 | 0.5839 | **0.5999** | 0.6311 | 0.6531 | **0.6687** | 0.5095 | 0.532 | **0.5426** |
| en | **0.7154** | 0.6932 | 0.7098 | 0.5771 | 0.6029 | **0.604** | 0.4733 | 0.5092 | **0.5143** |
| es | **0.6829** | 0.6649 | 0.6809 | 0.6328 | 0.6528 | **0.6626** | 0.5468 | 0.5634 | **0.5751** |
| hi | **0.6426** | 0.6155 | 0.6397 | 0.5529 | 0.6 | **0.6079** | 0.4425 | 0.4922 | **0.4949** |
| vi | **0.6405** | 0.6165 | 0.6397 | 0.573 | **0.6122** | 0.6069 | 0.4638 | **0.4908** | 0.4898 |
| zh | 0.662 | 0.628 | **0.6659** | 0.588 | **0.6352** | 0.6349 | 0.4668 | **0.5094** | 0.5081 |
| Mean | 0.6475 | 0.6256 | **0.6481** | 0.5843 | 0.62 | **0.6234** | 0.4731 | 0.5076 | **0.5121** |
| | R@1 | | | | | | R@10 | | |
| | Monolingual | | | Cross-lingual | | | Multilingual | | |
| Source Language | X-X-mono | X-Y | Proposed | X-X-mono | X-Y | Proposed | X-X-mono | X-Y | Proposed |
| ar | **0.4971** | 0.4778 | 0.4952 | 0.4142 | **0.4639** | 0.4583 | 0.528 | **0.5817** | 0.5811 |
| de | **0.4883** | 0.4785 | **0.498** | 0.5247 | 0.5394 | **0.5599** | 0.619 | 0.6462 | **0.6558** |
| en | **0.6307** | 0.6028 | 0.6237 | 0.4648 | 0.4916 | **0.4939** | 0.5833 | **0.6222** | 0.619 |
| es | 0.58 | 0.56 | **0.584** | 0.5174 | 0.5434 | **0.5587** | 0.651 | 0.6738 | **0.675** |
| hi | **0.5404** | 0.5168 | 0.5325 | 0.4306 | 0.4746 | **0.4821** | 0.5656 | 0.6187 | **0.6264** |
| vi | **0.544** | 0.5108 | **0.544** | 0.4536 | **0.4969** | 0.491 | 0.5752 | **0.6076** | 0.6058 |
| zh | 0.5437 | 0.5079 | **0.5556** | 0.4706 | 0.5193 | **0.5295** | 0.589 | **0.6417** | 0.6344 |
| Mean | 0.5463 | 0.5221 | **0.5476** | 0.468 | 0.5042 | **0.5105** | 0.5873 | 0.6274 | **0.6282** |

Table 5: Performance comparison of MAP and Recall scores across zero-shot monolingual, cross-lingual, and multilingual retrieval tasks on the XQuAD-R dataset for a fine-tuned LaBSE model and different training batch types. The best result is highlighted in **bold**, and the second-best result is underlined.

| Evaluation of Fine-tuned LaBSE Model on XQuAD-R Dataset | | | | | | | | |
|---|---|---|---|---|---|---|---|---|
| **MAP** | | | | | | | | |
| Monolingual | | | Cross-lingual | | | Multilingual | | |
| Source Language   X-X-mono | X-Y | Proposed | X-X-mono | X-Y | Proposed | X-X-mono | X-Y | Proposed |
| ar   0.7901 | 0.7848 | **0.7963** | 0.7257 | 0.7351 | **0.7356** | 0.6218 | **0.6481** | 0.6453 |
| de   0.8152 | 0.8135 | **0.8222** | 0.7667 | 0.774 | **0.7799** | 0.6632 | 0.6916 | **0.6945** |
| el   0.8022 | 0.7991 | **0.8121** | 0.7483 | 0.7603 | **0.762** | 0.6473 | **0.6783** | **0.6783** |
| en   0.8464 | 0.8349 | **0.8536** | 0.7932 | 0.7915 | **0.8074** | 0.6952 | 0.7183 | **0.7278** |
| es   0.812 | 0.8186 | **0.8331** | 0.7724 | 0.781 | **0.7892** | 0.6726 | 0.7021 | **0.7074** |
| hi   0.796 | 0.7824 | **0.8121** | 0.7382 | 0.7459 | **0.7582** | 0.6398 | 0.6625 | **0.6731** |
| ru   0.8243 | 0.8194 | **0.8314** | 0.7643 | 0.7745 | **0.7784** | 0.6684 | 0.6945 | **0.6948** |
| th   **0.7611** | 0.7371 | 0.7555 | 0.7123 | **0.7315** | 0.7294 | 0.6079 | **0.6377** | 0.6372 |
| tr   0.8086 | 0.794 | **0.8143** | 0.7541 | 0.7627 | **0.7691** | 0.655 | 0.6824 | **0.685** |
| vi   0.8136 | 0.8154 | **0.8285** | 0.7508 | 0.7646 | **0.7676** | 0.6506 | **0.6828** | 0.6809 |
| zh   0.8213 | 0.8096 | **0.8249** | 0.7451 | 0.759 | **0.7622** | 0.6464 | **0.672** | 0.6749 |
| Mean   0.8083 | 0.8008 | **0.8167** | 0.7519 | 0.7618 | **0.7672** | 0.6517 | 0.6791 | **0.6817** |
| **R@1** | | | | | | **R@10** | | |
| Monolingual | | | Cross-lingual | | | Multilingual | | |
| Source Language   X-X-mono | X-Y | Proposed | X-X-mono | X-Y | Proposed | X-X-mono | X-Y | Proposed |
| ar   0.7001 | 0.695 | **0.7127** | 0.6257 | 0.6349 | **0.6367** | 0.5438 | 0.5657 | **0.5671** |
| de   0.7293 | 0.7276 | **0.7386** | 0.6695 | 0.6784 | **0.6861** | 0.5742 | 0.6074 | **0.609** |
| el   0.7162 | 0.7137 | **0.7255** | 0.6517 | 0.6649 | **0.668** | 0.5673 | 0.5918 | **0.5967** |
| en   0.77 | 0.7582 | **0.7784** | 0.6996 | 0.6983 | **0.7189** | 0.6023 | 0.6308 | **0.6348** |
| es   0.7266 | 0.7401 | **0.7603** | 0.6752 | 0.6889 | **0.699** | 0.5828 | 0.6176 | **0.6186** |
| hi   0.7025 | 0.6805 | **0.721** | 0.6396 | 0.6469 | **0.6623** | 0.5599 | 0.58 | **0.5905** |
| ru   0.7445 | 0.7378 | **0.7538** | 0.6636 | 0.677 | **0.6832** | 0.5823 | **0.6088** | 0.6066 |
| th   **0.6703** | 0.6331 | 0.661 | 0.6108 | **0.6326** | 0.632 | 0.5322 | 0.5571 | **0.5594** |
| tr   0.7221 | 0.701 | **0.728** | 0.6561 | 0.6679 | **0.6733** | 0.5672 | 0.5971 | **0.5974** |
| vi   0.7276 | 0.7318 | **0.7487** | 0.6526 | 0.669 | **0.6732** | 0.5661 | **0.5979** | 0.5964 |
| zh   0.7392 | 0.718 | **0.7409** | 0.6452 | 0.6607 | **0.6684** | 0.5624 | 0.5882 | **0.5927** |
| Mean   0.7226 | 0.7124 | **0.7335** | 0.6536 | 0.6654 | **0.6728** | 0.5673 | 0.5948 | **0.5972** |

Table 6: Performance comparison of MAP and Recall scores across zero-shot monolingual, cross-lingual, and multilingual retrieval tasks on the MLQA-R dataset for a fine-tuned LaBSE model and different training batch types. The best result is highlighted in **bold**, and the second-best result is underlined.

| Evaluation of Fine-tuned LaBSE Model on MLQA-R Dataset | | | | | | | | |
|---|---|---|---|---|---|---|---|---|
| **MAP** | | | | | | | | |
| Monolingual | | | Cross-lingual | | | Multilingual | | |
| Source Language   X-X-mono | X-Y | Proposed | X-X-mono | X-Y | Proposed | X-X-mono | X-Y | Proposed |
| ar   **0.6293** | 0.6122 | 0.6283 | 0.6253 | 0.638 | **0.6441** | 0.5024 | **0.5271** | 0.5206 |
| de   0.6335 | 0.625 | **0.6405** | 0.6955 | 0.7095 | **0.7153** | 0.5756 | 0.5967 | **0.6013** |
| en   0.7347 | 0.7302 | **0.751** | 0.6534 | 0.6668 | **0.6733** | 0.5558 | 0.5787 | **0.5862** |
| es   **0.7186** | 0.7052 | 0.7106 | 0.6912 | 0.7073 | **0.709** | 0.6037 | 0.6205 | **0.6235** |
| hi   0.6783 | 0.6894 | **0.694** | 0.6478 | 0.6707 | **0.6883** | 0.5517 | 0.5792 | **0.5885** |
| vi   0.6699 | 0.663 | **0.6883** | 0.626 | **0.6521** | 0.6465 | 0.5258 | 0.5517 | **0.5573** |
| zh   **0.7009** | 0.6722 | 0.6924 | 0.6538 | **0.6926** | 0.6914 | 0.5375 | **0.5743** | 0.5721 |
| Mean   0.6807 | 0.671 | **0.6864** | 0.6561 | 0.6767 | **0.6811** | 0.5504 | 0.5755 | **0.5785** |
| **R@1** | | | | | | **R@10** | | |
| Monolingual | | | Cross-lingual | | | Multilingual | | |
| Source Language   X-X-mono | X-Y | Proposed | X-X-mono | X-Y | Proposed | X-X-mono | X-Y | Proposed |
| ar   **0.53** | 0.5106 | 0.5261 | 0.5145 | 0.5185 | **0.5359** | 0.6152 | **0.6438** | 0.6341 |
| de   0.5352 | 0.5234 | **0.5391** | 0.593 | 0.6021 | **0.6158** | 0.6886 | **0.7153** | **0.7153** |
| en   0.6376 | 0.6324 | **0.6672** | 0.546 | 0.5564 | **0.5682** | 0.6773 | 0.6976 | **0.6987** |
| es   **0.618** | 0.6 | 0.602 | 0.5844 | 0.6012 | **0.6007** | 0.7263 | 0.7325 | **0.7358** |
| hi   0.5779 | 0.5878 | **0.6036** | 0.5371 | 0.5572 | **0.5845** | 0.6788 | 0.7081 | **0.7097** |
| vi   0.5636 | 0.5577 | **0.591** | 0.5054 | **0.542** | 0.5318 | 0.6523 | 0.668 | **0.6691** |
| zh   **0.6071** | 0.5556 | 0.5873 | 0.5412 | 0.5853 | **0.5907** | 0.6572 | **0.7002** | 0.6959 |
| Mean   0.5813 | 0.5668 | **0.588** | 0.5459 | 0.5661 | **0.5754** | 0.6708 | **0.6951** | 0.6941 |

**Zero-shot Generalization to unseen languages.** The proposed approach exhibits remarkable zero-shot generalizability, as evidenced by its strong performance across different multilingual pre-trained models and evaluation datasets in Greek (el) and Vietnamese (vi) languages, which were not included in the training data used to develop the model. For example, in Table 5, which presents results for the LaBSE model on the XQuAD-R dataset, the proposed method achieves the best MAP and Recall@1 scores for Vietnamese, a low-resource language, in both monolingual and cross-lingual retrieval settings, outperforming the X-X-mono and X-Y approaches. In the multilingual retrieval setting, the proposed approach achieves MAP and R@10 scores of 0.6809 and 0.5964, respectively. These scores are very close to the 0.6828 and 0.5979 achieved by the X-Y model, which is primarily optimized for multilingual retrieval. Additionally, the proposed method significantly outperforms the X-X-mono approach, which is mainly optimized for monolingual retrieval and achieves scores of 0.6506 and 0.5661.

### 4.2.2 LANGUAGE BIAS EVALUATION

Tables 7 and 8 present a comprehensive comparison of the average rank distance metric[6] (Huang et al., 2023a) across different multilingual retrieval tasks using fine-tuned XLM-R and LaBSE models. The proposed approach is evaluated against two baseline methods: X-X-mono and X-Y, on two datasets: XQuAD-R (Table 7) and MLQA-R (Table 8). The lower the average rank distance, the better the performance.

**Significant mitigation of language bias Compared to monolingual batching.** The proposed approach substantially reduces language bias compared to the X-X-mono baseline. In Table 1, the proposed method achieves a mean rank distance of 286.6 using XLM-R, compared to 410.2 for X-X-mono, representing a 30.1% reduction in language bias. Similarly, for LaBSE, the proposed approach reduces the mean rank distance by 15.4% (from 261.5 to 221.1). In Table 2 (MLQA-R), the proposed method achieves a mean rank distance of 227.1 using XLM-R, compared to 287.5 for X-X-mono, resulting in a 21% reduction in language bias. For LaBSE, the proposed approach reduces the mean rank distance by 13.4% (from 225.3 to 195). These significant reductions highlight the effectiveness of the proposed method in mitigating language bias of the retrieval system.

**Competitive reduction in average rank distance compared to cross-lingual batching.** The proposed approach exhibits competitive performance in reducing the average rank distance compared to the strong X-Y baseline. In Table 7 (XQuAD-R), the proposed method achieves the best mean rank distance of 286.6 using XLM-R, outperforming both X-X-mono (295.4) and X-Y (295.4) baselines. For LaBSE, the proposed approach obtains a mean rank distance of 221.1, which is better than the X-Y baseline (225.2). In Table 8 (MLQA-R), the proposed method achieves a slightly higher mean rank distance than the X-Y baseline for XLM-R (227.1 vs. 226.7), but outperforms the X-Y baseline for LaBSE (195 vs. 198.3). These results demonstrate that the proposed approach is highly competitive in reducing the average rank distance and can even outperform the strong X-Y baseline in certain cases. This reduction in average rank distance directly translates to a decrease in language bias, as the proposed method effectively brings relevant documents closer together in the retrieval results, regardless of the language.

## 5 CONCLUSION

Developing IR models that can handle queries and documents across many languages is increasingly critical. In this work, we introduced a hybrid batch training strategy to optimize IR systems for monolingual, cross-lingual, and multilingual performance simultaneously. By fine-tuning multilingual language models on a mix of monolingual and cross-lingual question-answer pairs, the models learn robust representations that generalize well across languages and retrieval settings. Extensive experiments demonstrate that this simple yet effective approach consistently matches or outperforms models trained with only monolingual or cross-lingual data, and substantially mitigates the language bias that hinders multilingual retrieval performance.

---

[6]Rank distance is the average, over all queries and their relevant documents, of the difference between the maximum and minimum ranks assigned by an MLIR model to parallel (semantically similar) relevant documents across different languages.

## 6 LIMITATIONS

This work focuses on optimizing retrieval performance but does not address issues related to result diversity, fairness, or transparency in multilingual settings. For example, it may reflect societal biases present in the training data. Addressing these concerns is important for building equitable multilingual retrieval systems.

Furthermore, the experiments focus only on the XQuAD-R, MLQA-R, and MIRACL benchmark datasets. While these cover a range of languages, they may not be fully representative of real-world multilingual information retrieval needs. The robustness of the results to other domains, question types, and retrieval scenarios is an exciting future direction.

Table 7: Comparison of the rank distances among relevant documents of the XQuAD-R dataset across rank lists generated by fine-tuned XLM-R and LaBSE models for zero-shot multilingual retrieval tasks under different training batch types. The best result is highlighted in **bold**, and the second-best result is underlined.

| Average Rank Distance over XQuAD-R Dataset | | | | | | |
|---|---|---|---|---|---|---|
| | XLM-R | | | LaBSE | | |
| Source Language | X-X-mono | X-Y | Proposed | X-X-mono | X-Y | Proposed |
| ar | 552.8 | **371.5** | 376 | 332.4 | **279** | 285.4 |
| de | 356.6 | 252.8 | **242.1** | 214.9 | 192 | **175.1** |
| el | 431.6 | **307.8** | 311.9 | 251.3 | **224.4** | 228.4 |
| en | 320 | 239.6 | **219** | 189.3 | 162.1 | **150** |
| es | 371.4 | **264.5** | 267 | 235.4 | 210 | **188** |
| hi | 505.6 | 368.5 | **351.7** | 299.8 | **250.8** | 255.6 |
| ru | 367.9 | 271.7 | **245.6** | 226.5 | 195.5 | **189.3** |
| th | 431.6 | 316.9 | **304.4** | 391.5 | 325.9 | **323.9** |
| tr | 422.4 | 309 | **288.4** | 253.8 | 225.4 | **222.9** |
| vi | 395 | **289.4** | 295.6 | 245.2 | 208.6 | **204.8** |
| zh | 357.3 | 258.1 | **251.2** | 236.3 | **203.9** | 209 |
| Mean | 410.2 | 295.4 | **286.6** | 261.5 | 225.2 | **221.1** |

Table 8: Comparison of the rank distances among relevant documents of the MLQA-R dataset across rank lists generated by fine-tuned XLM-R and LaBSE models for zero-shot multilingual retrieval tasks under different training batch types. The best result is highlighted in **bold**, and the second-best result is underlined.

| Average Rank Distance over MLQA-R Dataset | | | | | | |
|---|---|---|---|---|---|---|
| | XLM-R | | | LaBSE | | |
| Source Language | X-X-mono | X-Y | Proposed | X-X-mono | X-Y | Proposed |
| ar | 298.2 | 248.1 | **247** | 245.7 | 223.5 | **208.9** |
| de | 248.4 | 219.7 | **211.5** | 204.1 | **179.9** | 194.7 |
| en | 458.4 | 371.6 | **366.9** | 340.6 | 304 | **291.3** |
| es | 179.7 | **146.7** | 135 | 152.6 | 145 | **143.6** |
| hi | 275 | 200.1 | **199** | 204.8 | 186.1 | **160.6** |
| vi | 296.6 | **213.2** | 223.4 | 225.2 | **194.6** | 205.5 |
| zh | 255.9 | **187.4** | 207.2 | 204.4 | **155.1** | 160.7 |
| Mean | 287.5 | **226.7** | 227.1 | 225.3 | 198.3 | **195** |

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

# A    APPENDIX

We provide additional information and detailed experimental results to support the main findings discussed in the body of the manuscript. It is organized into three main parts: (A.1) a description of the training datasets used to fine-tune the multilingual models, (A.2) an overview of the evaluation datasets and their characteristics, and (A.3) supplementary experimental results.

## A.1    TRAINING DATASETS

We present an overview of the training datasets used to fine-tune the multilingual pre-trained models. These datasets were selected to cover a diverse range of domains, tasks, and languages. These datasets vary in size, language coverage, and domain. The datasets mMARCO, Mr. Tydi, and MFAQ focus on multilingual tasks, while others like Google NQ, DuoRC, and NewsQA are monolingual. The datasets cover different domains, such as web search queries (Google NQ, WikiQA), movie plots (DuoRC), news articles (NewsQA), and FAQs (MFAQ).

- **DuoRC**: A paraphrased reading comprehension dataset aimed at evaluating complex language understanding. It contains over 186K question-answer pairs created from 7680 pairs of movie plot summaries (Saha et al., 2018).
- **EntityQuestions**: A dataset designed to challenge dense retrievers with simple entity-centric questions. It contains over 14K questions that require retrieving relevant entities from Wikipedia (Sciavolino et al., 2021).
- **Google NQ**: A QA dataset consisting of aggregated queries from Google's search engine, with annotated answers from Wikipedia pages. It contains over 300K queries and can be used for open-domain QA research (Kwiatkowski et al., 2019).
- **MFAQ**: A multilingual FAQ dataset containing over 100K question-answer pairs from 21 languages, covering topics like COVID-19, climate change, and more. It can be used for multilingual FAQ retrieval tasks (De Bruyn et al., 2021).
- **mMARCO**: A multilingual version of the MS MARCO passage ranking dataset, containing over 500K parallel queries and 9M passages in 13 languages. It can be used for multilingual information retrieval research (Bonifacio et al., 2021).
- **Mr. Tydi**: A multi-lingual benchmark for dense retrieval, consisting of monolingual and bilingual topic-document annotations in 11 languages. It's designed to evaluate the performance of multilingual dense retrieval models (Zhang et al., 2021).
- **NewsQA**: A machine comprehension dataset containing over 100K question-answer pairs based on CNN articles, aiming to encourage research on question answering from news articles (Trischler et al., 2017).
- **WikiQA**: An open-domain QA dataset with over 3K questions collected from Bing query logs, paired with answers extracted from Wikipedia. It's designed to be a challenge dataset for open-domain QA research (Yang et al., 2015).
- **Yahoo QA**: A dataset mined from Yahoo Answers, a QA website containing pairs of questions and answers.

Table 9 presents the dataset sizes after applying our in-house data processing pipeline to filter and clean the data. To expand the training data and cover a diverse set of languages, we employed an in-house machine translation pipeline (Fan et al., 2021; Kim et al., 2021; Costa-jussà et al., 2022). This pipeline was used to create parallel question-answer pairs across nine languages for the following monolingual datasets: WikiQA, DuoRC, NewsQA, Google NQ, Yahoo QA, and EntityQuestions. For the multilingual datasets, namely Mr. Tydi and MFAQ, only the English version was used. Additionally, mMARCO (Bonifacio et al., 2021), a multilingual version of the MS MARCO dataset, was included in the training data.

Table 9: Training data statistics.

| Dataset Name | Size per Language | Languages |
|---|---|---|
| WikiQA | 1,469 | en, ar, zh, de, es, ru, th, tr, hi |
| Mr. Tydi | 3,547 | en |
| DuoRC | 33,298 | en, ar, zh, de, es, ru, th, tr, hi |
| NewsQA | 59,496 | en, ar, zh, de, es, ru, th, tr, hi |
| Google NQ | 113,535 | en, ar, zh, de, es, ru, th, tr, hi |
| Yahoo QA | 135,557 | en, ar, zh, de, es, ru, th, tr, hi |
| EntityQuestions | 176,975 | en, ar, zh, de, es, ru, th, tr, hi |
| MFAQ | 3,567,659 | en |
| mMARCO | 39,780,811 | en, ar, zh, de, es, ru, hi |

Table 10: The number of queries and candidate sentences for each language in XQuAD-R and MLQA-R.

| | XQuAD-R | | MLQA-R | |
|---|---|---|---|---|
| | #Queries | #Candidates | #Queries | #Candidates |
| ar | 1190 | 1222 | 517 | 2545 |
| de | 1190 | 1276 | 512 | 2362 |
| el | 1190 | 1234 | - | - |
| en | 1190 | 1180 | 1148 | 6264 |
| es | 1190 | 1215 | 500 | 1787 |
| hi | 1190 | 1244 | 507 | 2426 |
| ru | 1190 | 1219 | - | - |
| th | 1190 | 852 | - | - |
| tr | 1190 | 1167 | - | - |
| vi | 1190 | 1209 | 511 | 2828 |
| zh | 1190 | 1196 | 504 | 2322 |

## A.2 EVALUATION DATASETS

We provide a summary of the evaluation datasets employed for conducting a zero-shot evaluation of the models developed in this work. It should be noted that these evaluation datasets were not used during the training phase of the models.

- **XQuAD-R** and **MLQA-R**: Two multilingual answer retrieval datasets derived from XQuAD (Artetxe et al., 2020; Rajpurkar et al., 2016) and MLQA (Lewis et al., 2020). They are designed to evaluate the performance of language-agnostic answer retrieval models. XQuAD-R is an 11-way parallel dataset where each question appears in 11 different languages and has 11 parallel correct answers across the languages. MLQA-R, on the other hand, covers 7 languages and has a variable number (2–4) of parallel correct answers across the corpus, with contexts surrounding the answer sentence not guaranteed to be parallel (Roy et al., 2020).

- **MIRACL dev**: A multilingual information retrieval dataset that covers a continuum of languages, featuring 18 languages with varying amounts of training data. It is designed to evaluate the performance of multilingual information retrieval models in low-resource settings and to facilitate research on cross-lingual transfer learning (Zhang et al., 2022).

Table 10 presents the number of questions and candidate sentences for each language in the XQuAD-R and MLQA-R datasets, while Table 11 displays the corresponding information for each language in the MIRACL Dev dataset.

Table 11: The number of queries and candidate sentences for each language in MIRACL Dev dataset.

| MIRACL Dev | | |
|---|---|---|
| Language | # Queries | # Candidates |
| ar | 2,869 | 2,061,414 |
| bn | 411 | 297,265 |
| en | 648 | 32,893,221 |
| es | 799 | 10,373,953 |
| fa | 632 | 2,207,172 |
| fi | 1,271 | 1,883,509 |
| fr | 343 | 14,636,953 |
| hi | 350 | 506,264 |
| id | 960 | 1,446,315 |
| ja | 860 | 6,953,614 |
| ko | 213 | 1,486,752 |
| ru | 1,252 | 9,543,918 |
| sw | 482 | 131,924 |
| te | 828 | 518,079 |
| th | 733 | 542,166 |
| zh | 393 | 4,934,368 |

## A.3 SUPPLEMENTARY EXPERIMENTAL RESULTS

We present additional experimental findings that complement the main results discussed in the paper. More specifically, we present zero-shot monolingual retrieval evaluation on the MIRACL dataset, showcasing the proposed approach's performance on a diverse set of languages. These supplementary results offer a more comprehensive understanding of the effectiveness of the proposed method and its ability to generalize across various retrieval tasks and languages.

### A.3.1 ZERO-SHOT MONOLINGUAL RETRIEVAL EVALUATION ON MIRACL

Tables 12 and 13 present the performance evaluation of fine-tuned XLM-R and LaBSE models on the MIRACL Dev dataset for zero-shot monolingual retrieval tasks across 15 languages. The models are evaluated using nDCG@10 and Recall@100 metrics, and the results are compared for three different training batch types: X-X-mono, X-Y, and the proposed hybrid batching approach.

When analyzing the performance of the XLM-R model, as shown in Table 12, the proposed approach achieves the second-best results in most cases for both nDCG@10 and Recall@100, often closely following the best-performing X-X-mono batch type. In some instances, such as for the Finnish, Russian, and French languages, the proposed method even surpasses the X-X-mono performance in terms of nDCG@10. Similarly, for languages like Persian, Japanese, and Spanish, the proposed approach outperforms X-X-mono in terms of Recall@100. Turning to the LaBSE model, presented in Table 13, the proposed approach frequently obtains the second-best results in both metrics and occasionally outperforms the X-X-mono batch type. This is particularly evident for the French, Chinese, Hindi, and Spanish languages in terms of nDCG@10, and for Chinese and Persian in terms of Recall@100.

For both XLM-R (Table 12) and LaBSE (Table 13) models, the proposed approach achieves higher mean and median scores compared to the X-Y batch type in nDCG@10 and Recall@100 metrics, indicating its superior overall performance. Although the X-X-mono batch type generally outperforms the proposed approach in terms of mean scores for both models and metrics, it is important to note that X-X-mono is specifically designed to optimize monolingual retrieval only. In contrast, the proposed hybrid batching approach is optimized for both monolingual and cross-lingual/multilingual retrieval.

Table 12: Performance comparison of nDCG and Recall scores across zero-shot monolingual retrieval tasks on the MIRACL Dev dataset for a fine-tuned XLM-R model and different training batch types. The best result is highlighted in **bold**, and the second-best result is underlined.

| Evaluation of Fine-tuned XLM-R Model on MIRACL Dev Dataset | | | | | | |
|---|---|---|---|---|---|---|
| | nDCG@10 | | | Recall@100 | | |
| Source Language | X-X-mono | X-Y | Proposed | X-X-mono | X-Y | Proposed |
| sw | 0.3319 | **0.3531** | 0.3348 | 0.6478 | **0.6503** | 0.6416 |
| bn | **0.5082** | 0.4442 | 0.4972 | **0.8738** | 0.8114 | 0.8621 |
| hi | **0.4144** | 0.3758 | 0.4071 | **0.7863** | 0.741 | 0.7706 |
| ko | **0.4364** | 0.4098 | 0.4261 | **0.7881** | 0.7204 | 0.783 |
| th | **0.5351** | 0.5072 | 0.5116 | **0.8727** | 0.8655 | 0.8564 |
| te | **0.5407** | 0.4511 | 0.4843 | **0.8671** | 0.7937 | 0.8366 |
| fi | 0.4658 | **0.5154** | 0.4791 | 0.8119 | **0.845** | 0.8224 |
| ja | **0.4294** | 0.4016 | 0.4189 | 0.7987 | 0.7786 | **0.804** |
| es | 0.2994 | **0.3098** | 0.2989 | 0.62 | 0.6237 | **0.624** |
| fr | 0.273 | **0.3044** | 0.2833 | 0.6968 | **0.7171** | 0.6674 |
| ru | 0.3317 | **0.3669** | 0.3444 | 0.6763 | **0.7169** | 0.6862 |
| zh | **0.3873** | 0.3438 | 0.3627 | **0.7983** | 0.7465 | 0.797 |
| fa | **0.4113** | 0.37 | 0.3937 | 0.786 | 0.7512 | **0.7958** |
| ar | **0.5403** | 0.4998 | 0.5203 | **0.8693** | 0.8152 | 0.8629 |
| id | 0.317 | **0.3363** | 0.3185 | 0.631 | **0.6539** | 0.6327 |
| Mean | **0.4148** | 0.3993 | 0.4054 | **0.7683** | 0.7487 | 0.7628 |
| Median | **0.4144** | 0.3758 | 0.4071 | 0.7881 | 0.7465 | **0.7958** |

Table 13: Performance comparison of nDCG and Recall scores across zero-shot monolingual retrieval tasks on the MIRACL Dev dataset for a fine-tuned LaBSE model and different training batch types. The best result is highlighted in **bold**, and the second-best result is underlined.

| Evaluation of Fine-tuned LaBSE Model on MIRACL Dev Dataset | | | | | | |
|---|---|---|---|---|---|---|
| | nDCG@10 | | | Recall@100 | | |
| Source Language | X-X-mono | X-Y | Proposed | X-X-mono | X-Y | Proposed |
| sw | **0.5076** | 0.4883 | 0.4896 | **0.8561** | 0.8177 | 0.8265 |
| bn | **0.5598** | 0.5155 | 0.5337 | **0.9194** | 0.8881 | 0.9048 |
| hi | 0.4325 | 0.3999 | **0.4381** | **0.7961** | 0.7655 | 0.7959 |
| ko | **0.4589** | 0.3963 | 0.4386 | **0.8253** | 0.7441 | 0.7903 |
| th | **0.5738** | 0.5285 | 0.5449 | **0.9013** | 0.8591 | 0.8585 |
| te | **0.5658** | 0.5013 | 0.5343 | **0.8768** | 0.8366 | 0.8458 |
| fi | **0.5327** | 0.506 | 0.5062 | **0.8631** | 0.8387 | 0.8303 |
| ja | **0.4333** | 0.3834 | 0.4027 | **0.822** | 0.7574 | 0.7884 |
| es | 0.3366 | 0.323 | **0.3396** | **0.6914** | 0.6594 | 0.6821 |
| fr | 0.3042 | 0.3124 | **0.3317** | **0.7472** | 0.7444 | 0.7448 |
| ru | **0.3839** | 0.3541 | 0.363 | **0.7421** | 0.7091 | 0.7132 |
| zh | 0.3768 | 0.3431 | **0.3912** | 0.7651 | 0.7628 | **0.7925** |
| fa | **0.4252** | 0.3777 | 0.4116 | 0.8103 | 0.7815 | **0.8189** |
| ar | **0.5783** | 0.5114 | 0.5391 | **0.8951** | 0.8403 | 0.8733 |
| id | **0.3572** | 0.3357 | 0.3522 | **0.6688** | 0.648 | 0.6656 |
| Mean | **0.4551** | 0.4184 | 0.4411 | **0.8120** | 0.7768 | 0.7954 |
| Median | 0.4333 | 0.3963 | **0.4381** | **0.822** | 0.7655 | 0.7959 |

