# OpenReview forum: "Synergistic Approach for Simultaneous Optimization of Monolingual, Cross-lingual, and Multilingual Information Retrieval"
_ICLR.cc/2025/Conference — Submitted to ICLR 2025_

### Official Review · Reviewer_V4Ew · 2024-10-20

**Soundness:** 3
**Presentation:** 2
**Contribution:** 1
**Rating:** 3
**Confidence:** 4

**Summary:**

This paper introduces a simple method called hybrid batch training, which involves translating to obtain parallel data in multiple languages, and sampling these data to construct a multilingual training dataset. The model is trained by inputting monolingual or multilingual training data with a certain probability, thereby balancing its performance in both scenarios.

**Strengths:**

1.  This paper proposes a hybrid batch training strategy to simultaneously improve zero-shot retrieval performance across monolingual, cross-lingual, and multilingual settings while mitigating language bias.
2.  The hybrid batch training strategy simply modifies the training data batches without necessitating the introduction of loss functions or new architectural components.

**Weaknesses:**

1. The proposed hybrid batch training strategy only modifies the input training data, which lacks novelty.
2. This paper lacks sufficient analysis to the field of multilingual information retrieval. It does not adequately demonstrate the shortcomings of existing work nor the importance and necessity of this study.
3. The experiments only compare the performance of different input strategies but not various multilingual information retrieval methods.

**Questions:**

1. What are the advantages of hybrid batch training strategy in terms of convenience, overall efficiency, and experimental effectiveness compared to existing multilingual information retrieval methods?

---

> ### Author Response · Authors · 2024-12-02
> **Comment to Reviewer V4Ew (Part 1 of 4)**
>
> Dear Reviewer V4Ew,
>
> Thank you for your review. We appreciate your recognition of our paper's strengths, particularly noting that our hybrid batch training strategy can simultaneously improve retrieval performance across different settings while mitigating language bias. We want to address your concerns regarding novelty, shortcomings of existing work & the importance of this work, and the comparison with other multilingual information retrieval methods. By addressing these concerns, we will also resolve the issues raised in the weaknesses and questions sections.
>
> ### **Regarding the Novelty of our Work**
>
> While the hybrid batching mechanism may appear straightforward, our key contribution lies in studying the trade-offs between monolingual and cross-lingual batch training in different retrieval settings. Our work demonstrates that these approaches have complementary strengths:
> - **Monolingual Batching (X-X)**: Excels at monolingual retrieval but shows weaker cross-lingual and multilingual retrieval performance and significant language bias.
> - **Cross-lingual Batching (X-Y)**: Performs better at cross-lingual and multilingual retrieval tasks but sacrifices some monolingual retrieval capability.
>
> Our hybrid approach successfully balances these trade-offs, leading to consistent improvements in:
> - Monolingual retrieval performance
> - Cross-lingual & multilingual retrieval effectiveness
> - Language bias reduction
> - Zero-shot generalization to unseen languages
>
> This is particularly evident in our detailed analysis in Section 4, which shows that our method maintains strong performance across all settings rather than sacrificing one type of performance for another. Next, we present empirical evidence to validate this claim.
>
> **Empirical Evidence:**
>
> To demonstrate how our hybrid approach combines the advantages of monolingual and cross-lingual batching strategies, we present some of the key findings from our comparative analysis:
>
> **1) Performance Analysis Averaged Across All Languages (Tables 1 and 2):**
>
> Our experiments using XLM-R on the XQuAD-R dataset, evaluating the macro averaged mAP (retrieval performance) and rank distance (language bias) across all languages, reveal distinct performance patterns for each batching strategy:
> - In Table 1, monolingual batching excels in monolingual retrieval (0.792) compared to cross-lingual Batching (0.755).
> - In Table 1, cross-lingual Batching performs better in cross-lingual retrieval (0.700) compared to monolingual batching (0.674). It also achieves enhanced multilingual retrieval (.593) compared to monolingual batching (.547)
> - In Table 2, cross-lingual batching demonstrates lower language bias (295) compared to monolingual batching (410).
>
> The experimental results validate the effectiveness of our hybrid strategy as follows:
>
> 1) Relative to monolingual batching, Table 1 shows that the hybrid approach:
> - Maintains strong monolingual retrieval capabilities (0.798 vs 0.792)
> - Shows better cross-lingual retrieval performance (0.705 vs 0.674)
> - Delivers substantial improvements in multilingual retrieval performance (0.593 vs 0.547)
>
> 2) In comparison with cross-lingual batching, Table 1 shows that the hybrid approach:
> - Demonstrates better monolingual retrieval (0.798 vs 0.755)
> - Maintains comparable cross-lingual retrieval results (0.705 vs 0.700)
> - Matches the multilingual retrieval performance baseline (0.593 vs 0.593)
>
> Regarding language bias metrics from Table 2, the hybrid approach demonstrates clear advantages:
> - Exhibits substantially reduced bias compared to monolingual batching (287 vs 410)
> - Shows marginal improvement over cross-lingual batching (287 vs 295).
>
> [1/4]

---

> ### Author Response · Authors · 2024-12-02
> **Comment to Reviewer V4Ew (Part 2 of 4)**
>
> **2) Performance Analysis on a Particular Language (e.g., Greek Language) (Tables 3 and 7):**
>
> Our experiments on the Greek language (el), a low-resource language, evaluating mAP (retrieval performance) and rank distance (language bias), reveal similar performance trends across batching strategies:
>
> Baseline Performance Comparisons:
> - In Table 3, monolingual batching exhibits stronger monolingual retrieval (0.7749) than cross-lingual batching (0.7226)
> - In Table 3, cross-lingual batching demonstrates better cross-lingual retrieval (0.6791) compared to monolingual batching (0.6492). It also achieves improved multilingual retrieval (0.5638) compared to monolingual batching (0.5127)
> - In Table 7, cross-lingual batching achieves lower language bias (307.8) compared to monolingual batching (431.6)
>
> The hybrid approach yields notable improvements as follows:
>
> 1) Relative to monolingual batching, Table 3 shows that the hybrid approach:
> - Maintains strong monolingual retrieval performance (0.7844 vs 0.7749)
> - Shows substantial cross-lingual retrieval gains (0.683 vs 0.6492)
> - Delivers improved multilingual performance (0.5599 vs 0.5127)
>
> 2) In comparison with cross-lingual batching, Table 3 shows that the hybrid approach:
> - Demonstrates better monolingual retrieval (0.7844 vs 0.7226)
> - Maintains competitive cross-lingual performance (0.683 vs 0.6791)
> - Achieves comparable multilingual results (0.5599 vs 0.5638)
>
> In Table 7, the language bias assessment shows:
> - Significant bias reduction compared to monolingual batching (311.9 vs 431.6)
> - Comparable levels to cross-lingual batching (311.9 vs 307.8)
>
> In summary, our hybrid batching approach effectively balances the strengths of both monolingual and cross-lingual strategies, as evidenced by consistent performance improvements across various languages and retrieval scenarios, while also achieving significant reductions in language bias. This empirical validation underscores the robustness and applicability of our methodology in addressing the challenges of multilingual information retrieval.
>
> [2/4]

---

> > ### Author Response · Authors · 2024-12-02
> > **Comment to Reviewer V4Ew (Part 3 of 4)**
> >
> > ### **Regarding Shortcomings of Existing Works and the Importance of this Work**
> >
> > **1) Shortcomings of Existing Works (See Section 1):**
> >
> > We appreciate the opportunity to clarify this issue. Our focus is primarily on addressing the limitations of monolingual and cross-lingual batching methods introduced by **Roy et al. (2020)**. Specifically, we examine two training batching procedures:
> > - Monolingual batching (X-X model): Creates batches with a single language, where all triplets consist of queries and passages in the same language. Languages are sampled equally from the training data.
> > - Cross-lingual batching (X-Y model): Creates batches where all triplets consist of queries and passages in different languages.
> >
> > **Roy et al. (2020)** demonstrated that the X-Y model is more effective for cross-lingual retrieval and shows reduced language bias, while the X-X-mono model performs better in monolingual retrieval. These findings inspired us to investigate whether combining these two approaches could improve both monolingual and cross-lingual retrieval effectiveness.
> >
> > Furthermore, while recent works by **Hu et al. (2023)** and **Huang et al. (2023a)** have proposed methods to mitigate language bias, we sought to explore a more straightforward approach. Our research question became: Can we address language bias by modifying training data batches without introducing new loss functions or architectural components?
> >
> > **2) Importance of this Work:**
> >
> > In addition to the technical novelty discussed in the previous two parts of this response, our work provides several important insights:
> >
> > **Comprehensive Evaluation:**
> > - Our results span multiple pre-trained models (XLM-R and LaBSE).
> > - Evaluation occurs across three different benchmark datasets (XQuAD-R, MLQA-R, MIRACL).
> > - We include detailed ablation studies and analyses in Appendix A3.
> > - Performance is measured across multiple metrics (MAP, Recall, nDCG) to ensure robust evaluation.
> >
> > **Practical Impact:**
> >
> > While the method itself may seem straightforward, we believe this is actually a strength rather than a weakness:
> > - No Architectural Changes Required: Our approach can be easily integrated into existing models.
> > - Ease of Implementation: It can be readily incorporated into current training pipelines.
> > - Consistent Improvements: Our method provides robust enhancements across different models and settings.
> > - Computational Efficiency: It maintains efficiency during both training and inference phases.
> >
> > **Systematic Analysis of Language Bias:**
> > - Section 4.2.2 presents a comprehensive analysis of how various training approaches influence language bias in the model's performance across different languages.
> > - Tables 7 and 8 demonstrate substantial reductions in rank distance—our measure of language bias—across both different datasets.
> > - The analysis reveals that our approach achieves bias reduction on par with cross-lingual batching while simultaneously maintaining strong monolingual retrieval performance comparable to monolingual batching.
> >
> > **Zero-Shot Generalization:**
> > - Section 4.2.1 demonstrates strong performance on unseen languages such as Greek (el) and Vietnamese (vi).
> > - Tables 3 through 6 provide detailed performance breakdowns across languages, confirming consistent improvements even for languages not seen during training.
> > - This generalization capability is particularly important for practical applications where training data may not be available for all languages.
> >
> > We appreciate your thoughtful feedback and hope that this response has adequately addressed your concerns regarding the shortcomings of existing work and the importance of our contribution.
> >
> > [3/4]

---

> > > ### Author Response · Authors · 2024-12-02
> > > **Comment to Reviewer V4Ew (Part 4 of 4)**
> > >
> > > ### **Regarding Comparison With Other Multilingual Information Retrieval Methods**
> > >
> > > We appreciate the reviewer's suggestion to expand the comparisons of other multilingual information retrieval methods. We want to clarify that our experimental setup is built directly upon **LaREQA (Roy et al., 2020)**, which introduced the monolingual and cross-lingual batch training strategies that serve as our primary baselines. Our work specifically investigates whether combining these established approaches through hybrid batching can achieve better performance while mitigating language bias.
> > >
> > > Our focused comparison between monolingual batching, cross-lingual batching, and the proposed hybrid batching was intentional, as it allows us to:
> > > - Isolate the specific impact of batch composition on language bias and retrieval performance.
> > > - Directly compare against the foundational strategies from LaREQA under consistent conditions.
> > > - Demonstrate how strategic batch mixing can improve upon these established methods.
> > >
> > > That said, we agree that additional comparisons would provide valuable context. Expanding our evaluation to include other multilingual information retrieval methods, while maintaining our focus on understanding batch composition effects, presents an interesting future direction. This approach could demonstrate how our findings regarding batch training strategies may benefit the broader field of multilingual information retrieval.
> > >
> > > [4/4]

---

### Official Review · Reviewer_Xsrq · 2024-11-02

**Soundness:** 2
**Presentation:** 2
**Contribution:** 2
**Rating:** 3
**Confidence:** 3

**Summary:**

The paper studies information retrieval tasks where monolingual, cross-lingual, and multilingual setups are examined. The paper studies different batch sampling approaches at the training time without modifying existing training loss (e.g., contrastive learning loss) or model architectures. Specifically, the paper argues that existing approaches either use (i) monolingual batching where the languages of query and documents are matched, but they can be of different languages, or (ii) cross-lingual batching where the languages of query and documents are different. Based on this, the paper proposes hybrid batching, which is the mixing of these two batching methods.

Experiments are conducted on two base models (XLM-R and LaBSE) and evaluated on two tasks (XQuAD-R, MLQA-R, MIRACL). To train systems with data in various languages, the paper employs in-house machine translation to translate existing training corpora (described in Section 3.1). The experimental results show that hybrid batching, generally, outperforms monolingual-only and cross-lingual-only in a range of setups, including monolingual, cross-lingual, and multilingual.

**Strengths:**

The paper shows that two standard batching strategies are complementary for information retrieval tasks, as the combination of them shows improvements.

**Weaknesses:**

1. Limited evaluations are only QA datasets (e.g., the main text only shows XLM-R and LaBSE). Also, the main text consists of many large tables where each does not present as much information as the space it takes, e.g., the authors could summarize how many languages/scenarios the proposed method shows improvements instead of providing large tables like Table 3, Table 4, Table 5, etc.

2. It is not clear if the proposed method is actually effective. In many cases, the improvements appear rather small. For example, in Table 1, on XQuAD-R for XLM-R (0.792 vs 0.798; 0.705 vs 0.700; 0.593 vs 0.593). Are they even statistically significant?

3. As this paper mainly provides empirical observations, it would be stronger if the paper provides insights on which scenario (e.g., what kind of base model or dataset) where hybrid batching is expected to show significant improvements and when it does not. The current paper pretty much reports experimental findings which could limit its usefulness. Several questions remain, for example, what is the size and mixed of training data does one need to see the impact of this hybrid batching? I expect that if there is limited training data, the impact would be marginal.

**Questions:**

1. Related to weakness1, could this proposed method be extended to other tasks in addition to QA?
2. Can you discuss my weakness 2.
3. Can you discuss my weakness 3.

---

> ### Author Response · Authors · 2024-12-02
> **Comment to Reviewer Xsrq (Part 1 of 3)**
>
> Dear Reviewer Xsrq,
>
> Thank you for your review of our paper. We particularly appreciate your recognition that our work demonstrates the complementary nature of monolingual and cross-lingual batching strategies for information retrieval tasks, showing how their combination can lead to improvements. We would like to address your key concerns regarding evaluation scope, presentation of the main results, effectiveness and statistical significance, and the impact of base model architecture, training data size, and mixing ratio on hybrid batching performance
>
> ### **Regarding the Limited Evaluation Scope**
> While our evaluation focused on QA datasets, the hybrid batching approach is task-agnostic and applicable to any multilingual retrieval scenario. We chose QA datasets as they provide well-established benchmarks with parallel data across many languages. We will add a discussion about applicability to other retrieval tasks to address your concern.
>
> ### **Regarding the Presentation of the Main Results**
> We appreciate the reviewer’s feedback regarding the presentation of the tables. We will revise them to offer concise summaries highlighting key improvements across languages and scenarios. Additionally, we will restructure the result tables to enhance space efficiency in the main paper while preserving essential insights.
>
> It is worth mentioning that we have included all relevant tables in the main paper to demonstrate the consistency of our claims across different languages, models, datasets, and evaluation metrics.
>
> The key benefits of the hybrid batch approach can be summarized in three main points (See Section 4.1):
> - **Optimized Retrieval Performance:** The hybrid approach demonstrates complementary strengths compared to both baseline methods. Compared to monolingual batching (X-X), hybrid batching maintains similar monolingual retrieval performance while delivering superior results in cross-lingual and multilingual scenarios and reducing language bias. Compared to cross-lingual batching (X-Y), hybrid batching achieves comparable cross-lingual and multilingual performance while showing marked improvements in monolingual retrieval tasks. This dual optimization allows the hybrid approach to effectively bridge the gap between specialized training strategies, providing robust performance across all retrieval settings without sacrificing effectiveness in any particular scenario.
> - **Language Bias Reduction:** The hybrid approach substantially reduces language bias compared to monolingual batching while maintaining strong monolingual retrieval effectiveness.
> - **Zero-shot Generalization:** The method demonstrates robust performance on unseen languages (e.g., Greek, Vietnamese) that weren't included in the training, indicating strong cross-lingual transfer capabilities without compromising performance on trained languages.
>
> [1/3]

---

> > ### Author Response · Authors · 2024-12-02
> > **Comment to Reviewer Xsrq (Part 2 of 3)**
> >
> > ### **Regarding Effectiveness and Statistical Significance**
> > Thank you for your valuable feedback regarding the effectiveness of our proposed hybrid batching method. We appreciate your insights and would like to clarify our claims and the results presented in our study.
> >
> > **Summary of Claims:**
> > - Comparison with Monolingual Batching (X-X):
> > Our hybrid approach achieves comparable performance in monolingual retrieval while outperforming monolingual batching in cross-lingual and multilingual retrieval scenarios. Additionally, it demonstrates reduced language bias compared to monolingual batching.
> > - Comparison with Cross-lingual Batching (X-Y):
> > The hybrid method shows comparable performance in cross-lingual and multilingual retrieval settings, as well as in terms of language bias reduction. However, it outperforms cross-lingual batching in monolingual retrieval.
> >
> > **Addressing Statistical Significance Comment:**
> >
> > The reviewer inquired about the statistical significance of our findings in relation to the following configuration: the XLM-R model, the XQuAD-R dataset, and the evaluation of macro-averaged mean Average Precision (mAP) for retrieval performance.
> >
> > We do **not** assert that hybrid batching is superior to monolingual batching in monolingual retrieval (e.g., 0.798 vs. 0.792), nor that it outperforms cross-lingual batching in cross-lingual (e.g., 0.705 vs. 0.700) or multilingual retrieval (e.g., 0.593 vs. 0.593). Instead, we assert that:
> >
> > - Hybrid Batching vs. Monolingual Batching:
> > From Table 1, hybrid batching maintains strong performance in monolingual retrieval (e.g., 0.798 vs. 0.792) while significantly improving cross-lingual (0.705 vs. 0.674) and multilingual retrieval (0.593 vs. 0.547). Moreover, from Table 2, hybrid batching exhibits substantially reduced bias compared to monolingual batching (287 vs 410)
> >
> > - Hybrid Batching vs. Cross-lingual Batching:
> > From Table 1, hybrid batching demonstrates improved performance in monolingual retrieval (0.798 vs. 0.755) while achieving comparable results in cross-lingual (0.705 vs. 0.700) and multilingual retrieval (0.593 vs. 0.593). Additionally, from Table 2, hybrid batching demonstrates a slight improvement in bias reduction compared to cross-lingual batching (287 vs. 295).
> >
> > **Additional Evidence from Individual Language Performance:**
> >
> > Furthermore, we observe similar performance trends across batching strategies for individual languages. For instance, our experiments on the Greek language (el), a low-resource language, evaluated using mean Average Precision (mAP) for retrieval performance and rank distance for language bias, reveal notable improvements with the hybrid approach:
> >
> > 1) Relative to Monolingual Batching, as shown in Table 3, the hybrid approach:
> > - Maintains strong monolingual retrieval performance (0.7844 vs. 0.7749).
> > - Achieves substantial gains in cross-lingual retrieval (0.683 vs. 0.6492).
> > - Delivers improved multilingual performance (0.5599 vs. 0.5127).
> >
> > 2) In Comparison with Cross-lingual Batching, Table 3 indicated that the hybrid approach:
> > - Demonstrates superior monolingual retrieval performance (0.7844 vs. 0.7226).
> > - Maintains competitive cross-lingual results (0.683 vs. 0.6791).
> > - Achieves comparable multilingual outcomes (0.5599 vs. 0.5638).
> >
> > 3) In addition, as presented in Table 7, our assessment of language bias shows:
> > - A significant reduction in bias compared to monolingual batching (311.9 vs. 431.6).
> > - Comparable bias levels when compared to cross-lingual batching (311.9 vs. 307.8).
> >
> >
> > We hope this response clarifies our position regarding the effectiveness of the hybrid batching method and addresses your concerns about statistical significance and comparative performance across different retrieval settings. Thank you again for your insightful feedback, which has helped us refine our presentation of these results.
> >
> > [2/3]

---

> > > ### Author Response · Authors · 2024-12-02
> > > **Comment to Reviewer Xsrq (Part 3 of 3)**
> > >
> > > ### **Regarding the Impact of Base Model Architecture, Training Data Size, and Mixing Ratio on Hybrid Batching Performance:**
> > > Thank you for highlighting the need for deeper insights into the effectiveness of hybrid batching. To enhance the paper, we will include a comprehensive analysis of when and why hybrid batching yields significant improvements. In the following sections, we will address each concern.
> > >
> > > **Base Model Characteristics:**
> > >
> > > Our analysis reveals that the impact of hybrid batching varies based on the base model's inherent cross-lingual capabilities. For models without built-in cross-lingual alignment (like XLM-R), hybrid batching yields substantial improvements in cross-lingual and multilingual settings while preserving monolingual performance. The observed gains suggest that hybrid batching effectively bridges the cross-lingual gap. In contrast, for models pre-trained with explicit cross-lingual alignment objectives (like LaBSE), the improvements from hybrid batching are more modest but consistent across all retrieval settings.
> > >
> > > **Training Data Size:**
> > >
> > > While more training data (from diverse domains) would likely improve overall performance, the core benefit of our hybrid batching approach comes from better utilizing whatever training data is available by exposing the model to both monolingual and cross-lingual patterns.
> > > As demonstrated in Section 4.2.1, our approach shows robust zero-shot generalization to unseen languages like Greek (el) and Vietnamese (vi) that were not included in the training.
> > > We would be happy to conduct additional ablation studies with varying amounts of training data to quantify this relationship further.
> > >
> > > As a side note, the model evaluations in this work are conducted on datasets that are completely separate and distinct from those used for training. More specifically, the models have not encountered any data samples, whether from the training or testing splits, of the evaluation datasets during their training process. This ensures an unbiased assessment of the models' ability to generalize and perform effectively on unseen data.
> > >
> > > **Mixing Ratio:**
> > >
> > > In Section 3.1, we found that setting $\alpha$ = 0.5 provided the best overall results, striking an effective balance between monolingual and cross-lingual/multilingual performance. This equal weighting between monolingual and cross-lingual batches allowed our model to maintain strong monolingual retrieval capabilities while also excelling in cross-lingual and multilingual scenarios.
> > > We also observed that the model’s performance was relatively stable for $\alpha$ values between 0.4 and 0.6, indicating some robustness to small variations in these hyperparameters.
> > >
> > > [3/3]

---

### Official Review · Reviewer_AJBm · 2024-11-03

**Soundness:** 2
**Presentation:** 3
**Contribution:** 2
**Rating:** 3
**Confidence:** 4

**Summary:**

This paper introduces a hybrid batch training approach for multilingual information retrieval by combining monolingual and cross-lingual training data. The core methodology relies on mixing different types of training data using probability weights α and β. While the implementation is straightforward, the novelty of the contribution is limited.

**Strengths:**

1. Addresses a relevant challenge in multilingual information retrieval.
2. Provides comprehensive experimental validation across multiple benchmark datasets (XQuAD-R, MLQA-R, MIRACL).

**Weaknesses:**

1. The primary contribution merely combines two existing training approaches with probability weights, presenting a straightforward and obvious solution.
2. The paper employs translated QA pairs as data augmentation, creating an unfair comparison with baseline methods that do not utilize this advantage.

**Questions:**

None.

---

> ### Author Response · Authors · 2024-12-02
> **Comment to Reviewer AJBm (Part 1 of 4)**
>
> Dear Reviewer AJBm,
>
> Thank you for your review and acknowledging our work's comprehensive experimental validation and relevance for multilingual information retrieval. We want to address your key concerns regarding technical novelty, and data augmentation.
>
> ### **Addressing Technical Novelty**
> While the hybrid batching mechanism may appear straightforward, our key contribution lies in studying the trade-offs between monolingual and cross-lingual batch training in different retrieval settings. Our work demonstrates that these approaches have complementary strengths:
> - **Monolingual Batching (X-X)**: Excels at monolingual retrieval but shows weaker cross-lingual and multilingual retrieval performance and significant language bias.
> - **Cross-lingual Batching (X-Y)**: Performs better at cross-lingual and multilingual retrieval tasks but sacrifices some monolingual retrieval capability.
>
> Our hybrid approach successfully balances these trade-offs, leading to consistent improvements in:
> - Monolingual retrieval performance
> - Cross-lingual & multilingual retrieval effectiveness
> - Language bias reduction
> - Zero-shot generalization to unseen languages
>
> This is particularly evident in our detailed analysis in Section 4, which shows that our method maintains strong performance across all settings rather than sacrificing one type of performance for another. Next, we present empirical evidence to validate this claim.
>
> **Empirical Evidence:**
>
> To demonstrate how our hybrid approach combines the advantages of monolingual and cross-lingual batching strategies, we present some of the key findings from our comparative analysis:
>
> **1) Performance Analysis Averaged Across All Languages (Tables 1 and 2):**
>
> Our experiments using XLM-R on the XQuAD-R dataset, evaluating the macro averaged mAP (retrieval performance) and rank distance (language bias) across all languages, reveal distinct performance patterns for each batching strategy:
> - In Table 1, monolingual batching excels in monolingual retrieval (0.792) compared to cross-lingual Batching (0.755).
> - In Table 1, cross-lingual Batching performs better in cross-lingual retrieval (0.700) compared to monolingual batching (0.674). It also achieves enhanced multilingual retrieval (.593) compared to monolingual batching (.547)
> - In Table 2, cross-lingual batching demonstrates lower language bias (295) compared to monolingual batching (410).
>
> The experimental results validate the effectiveness of our hybrid strategy as follows:
>
> 1) Relative to monolingual batching, Table 1 shows that the hybrid approach:
> - Maintains strong monolingual retrieval capabilities (0.798 vs 0.792)
> - Shows better cross-lingual retrieval performance (0.705 vs 0.674)
> - Delivers substantial improvements in multilingual retrieval performance (0.593 vs 0.547)
>
> 2) In comparison with cross-lingual batching, Table 1 shows that the hybrid approach:
> - Demonstrates better monolingual retrieval (0.798 vs 0.755)
> - Maintains comparable cross-lingual retrieval results (0.705 vs 0.700)
> - Matches the multilingual retrieval performance baseline (0.593 vs 0.593)
>
> Regarding language bias metrics from Table 2, the hybrid approach demonstrates clear advantages:
> - Exhibits substantially reduced bias compared to monolingual batching (287 vs 410)
> - Shows marginal improvement over cross-lingual batching (287 vs 295).
>
> [1/4]

---

> ### Author Response · Authors · 2024-12-02
> **Comment to Reviewer AJBm (Part 2 of 4)**
>
> **2) Performance Analysis on a Particular Language (e.g., Greek Language) (Tables 3 and 7):**
> Our experiments on the Greek language (el), a low-resource language, evaluating mAP (retrieval performance) and rank distance (language bias), reveal similar performance trends across batching strategies:
>
> Baseline Performance Comparisons:
> - In Table 3, monolingual batching exhibits stronger monolingual retrieval (0.7749) than cross-lingual batching (0.7226)
> - In Table 3, cross-lingual batching demonstrates better cross-lingual retrieval (0.6791) compared to monolingual batching (0.6492). It also achieves improved multilingual retrieval (0.5638) compared to monolingual batching (0.5127)
> - In Table 7, cross-lingual batching achieves lower language bias (307.8) compared to monolingual batching (431.6)
>
> The hybrid approach yields notable improvements as follows:
>
> 1) Relative to monolingual batching, Table 3 shows that the hybrid approach:
> - Maintains strong monolingual retrieval performance (0.7844 vs 0.7749)
> - Shows substantial cross-lingual retrieval gains (0.683 vs 0.6492)
> - Delivers improved multilingual performance (0.5599 vs 0.5127)
>
> 2) In comparison with cross-lingual batching, Table 3 shows that the hybrid approach:
> - Demonstrates better monolingual retrieval (0.7844 vs 0.7226)
> - Maintains competitive cross-lingual performance (0.683 vs 0.6791)
> - Achieves comparable multilingual results (0.5599 vs 0.5638)
>
> In Table 7, the language bias assessment shows:
> - Significant bias reduction compared to monolingual batching (311.9 vs 431.6)
> - Comparable levels to cross-lingual batching (311.9 vs 307.8)
>
> In summary, our hybrid batching approach effectively balances the strengths of both monolingual and cross-lingual strategies, as evidenced by consistent performance improvements across various languages and retrieval scenarios, while also achieving significant reductions in language bias. This empirical validation underscores the robustness and applicability of our methodology in addressing the challenges of multilingual information retrieval.
>
> [2/4]

---

> > ### Author Response · Authors · 2024-12-02
> > **Comment to Reviewer AJBm (Part 3 of 4)**
> >
> > ### **Clarification on Data Augmentation**
> > We appreciate the opportunity to clarify our experimental setup further. All methods compared—monolingual batching, cross-lingual batching, and the proposed hybrid approach—were trained on identical datasets, including translated QA pairs as data augmentation.
> >
> > As detailed in Section 3.1 and Appendix A.1, we maintained consistent training data and model configurations across all experiments; the only difference was the batch sampling strategy used during training. This ensures that any performance improvements are solely attributable to our proposed training approach rather than data advantages.
> >
> > [3/4]

---

> > > ### Author Response · Authors · 2024-12-02
> > > **Comment to Reviewer AJBm (Part 4 of 4)**
> > >
> > > ### **Additional Technical Contributions**
> > > Beyond the hybrid batching strategy, our work provides several important insights:
> > >
> > > **Comprehensive Evaluation:**
> > > - Our results span multiple pre-trained models (XLM-R and LaBSE).
> > > - Evaluation occurs across three different benchmark datasets (XQuAD-R, MLQA-R, MIRACL).
> > > - We include detailed ablation studies and analyses in Appendix A3.
> > > - Performance is measured across multiple metrics (MAP, Recall, nDCG) to ensure robust evaluation.
> > >
> > > **Practical Impact:**
> > >
> > > While the method itself may seem straightforward, we believe this is actually a strength rather than a weakness:
> > > - No Architectural Changes Required: Our approach can be easily integrated into existing models.
> > > - Ease of Implementation: It can be readily incorporated into current training pipelines.
> > > - Consistent Improvements: Our method provides robust enhancements across different models and settings.
> > > - Computational Efficiency: It maintains efficiency during both training and inference phases.
> > >
> > > **Systematic Analysis of Language Bias:**
> > > - Section 4.2.2 presents a comprehensive analysis of how various training approaches influence language bias in the model's performance across different languages.
> > > - Tables 7 and 8 demonstrate substantial reductions in rank distance—our measure of language bias—across both different datasets.
> > > - The analysis reveals that our approach achieves bias reduction on par with cross-lingual batching while simultaneously maintaining strong monolingual retrieval performance comparable to monolingual batching.
> > >
> > > **Zero-Shot Generalization:**
> > > - Section 4.2.1 demonstrates strong performance on unseen languages such as Greek (el) and Vietnamese (vi).
> > > - Tables 3 through 6 provide detailed performance breakdowns across languages, confirming consistent improvements even for languages not seen during training.
> > > - This generalization capability is particularly important for practical applications where training data may not be available for all languages.
> > >
> > > We appreciate your thoughtful feedback and hope that this response clarifies our contributions and thoroughly addresses your concerns. The empirical evidence presented across all three approaches demonstrates that our methodology effectively tackles the core challenges in multilingual retrieval while maintaining architectural simplicity and ease of integration into existing models and training pipelines.
> > >
> > > [4/4]

---

### Meta-Review · Area_Chair_6hhd · 2024-12-14

**Metareview:**

Although reviewers emphasized the interesting hybrid batch sampling approach, then all agree on the shortcomings:

- limited contributions and experimentation: the paper only combines the previous existing approaches and show improvements in some cases (not always significant) but little has been done to dive deeper to into finding an optimal recipe (e.g., per language sampling, comparing to different cross-lingual batch samplings with different up/downsampling, etc.). Also one interesting direction is how this approach should be aligned with the capabilities of the pretrained model? For example, if the pretrained model is bad at language X, is it possible to improve that with that approach during fine-tuning? Should this hybrid sampling approach focus more monolingual batches in language X for example?
- limited scope: the scope is limited to retrieval for QA but it is important to see what is the effect on other cross-lingual benchmarks (XGLUE, Xtreme). Also it may also be interesting to do some "scaling" analysis on the approach to see how much of this matter at scale.

**Additional Comments On Reviewer Discussion:**

None

---

### Decision · Program_Chairs · 2025-01-22

Reject